# Rheological Investigation of Relaxation Behavior of Polycarbonate/Acrylonitrile-Butadiene-Styrene Blends

**DOI:** 10.3390/polym12091916

**Published:** 2020-08-25

**Authors:** Jae Sik Seo, Ho Tak Jeon, Tae Hee Han

**Affiliations:** 1Department of Organic and Nano Engineering, Hanyang University, Seoul 04763, Korea; jaesik.seo@hyundai.com; 2Interior System Plastic Materials Development Team, Material Development Center, Hyundai Motor Company, Hwaseong 18280, Korea

**Keywords:** polycarbonate, acrylonitrile-butadiene-styrene, relaxation, rheology, polymer blend

## Abstract

The rheological properties of polycarbonate/acrylonitrile-butadiene-styrene (PC/ABS) blends with various blend ratios are investigated at different temperatures to determine the shear dependent chain motions in a heterogeneous blend system. At low frequency levels under 0.1 rad/s, the viscosity of the material with a blend ratio of 3:7 (PC:ABS) is higher than that of pure ABS polymer. As the temperature increases, the viscosities of ABS-rich blends increase rather than decrease, whereas PC-rich blends exhibit decrease in viscosity. Results from the time sweep measurements indicate that ordered structures of PC and the formation and breakdown of internal network structures of ABS polymer occur simultaneously in the blend systems. Newly designed sequence test results show that the internal structures formed between PC and ABS polymers are dominant at low shear conditions for the blend ratio of 3:7 and effects of structural change and the presence of polybutadiene (PBD) become dominant at high shear conditions for pure ABS. The results of yield stress and relaxation time for PC/ABS blends support this phenomenon. The specimen with a blend ratio of 3:7 exhibited the highest value of yield stress at high temperature among others, which implies that the internal structure become stronger at higher temperature. The heterogeneity of ABS-rich blends increases whereas that of PC-rich blends decreases as temperature increases.

## 1. Introduction

Polymer blending techniques are simple methods to address the weakness of various polymers. One of the most successful polymeric blends, polycarbonate (PC) and acrylonitrile-butadiene-styrene (ABS) blends, has been widely used in automotive industrial applications. The weaknesses of PC polymers (including notch sensitivity, low processability, thickness-dependent physical properties) can be overcome by incorporation of ABS polymer [1,2,3,4,5,6,7]. The presence of ABS polymer synergistically optimizes the characteristics of PC polymer, providing superior impact properties, toughness and tensile strength among other features. A number of studies have been carried out investigating blend ratios, molecular weights and amounts of butadiene rubber in ABS among other aspects [8,9,10,11,12,13,14,15]. Many investigations have been carried out on the morphologies of PC/ABS blends forming co-continuous phases between PC, styrene-co-acrylonitrile (SAN) and polybutadiene (PBD), which have significant effects on physical properties [16,17,18,19,20].

Most automotive parts are manufactured by injection molding of molten polymers. The dimensions of automotive interior or exterior parts are quite large and they may be complex so that determining the optimum processing conditions is extremely difficult. Thus, these conditions must be studied thoroughly prior to serial production. It is crucial to determine the process required to go from the molten material to the final product because automobiles are subjected to severe conditions such as heat, aging, humidity, UV radiation, vibration, pollution and corrosion during their service life. This indicates that even small defects during processing can directly influence the passenger’s safety. Hence, a knowledge of the rheological properties of materials is significantly essential for determining processing parameters. In addition, rheology can be applied to both the processing conditions and analysis the chain motion of polymer blends because bulky and systematic responses can be obtained by involving microscopic chain deformation at given controllable processing conditions such as shear rate, strain, temperature and time [21,22,23,24,25,26].

Numerous studies of immiscible PC/ABS polymer blends have focused on their physical properties [27,28,29,30,31,32,33,34,35,36]. The relationship between the main components, PC, SAN and vulcanized PBD, are extremely complicated and the chain relaxation of these tertiary blends are complex. The rheological properties of the immiscible polymer blends systems are closely related to the morphology formation and compatibility between the blend components [37,38,39]. Because the rheological responses are systematical and macroscopic, little differences in molecular motions can bring about different viscoelastic properties. Thus, there have been few papers focusing on the rheological properties of PC/ABS polymer blends regarding the complex chain motions. Conventional rheological measurements have been implemented in the small amplitude strain region in which polymer chains exhibit linear deformations under given shear conditions. Recent studies have focused on using large amplitude strain analysis techniques, in which microstructures formed as a result of the complex polymeric chains response show non-linear characteristics [40,41,42,43,44,45,46]. We investigated the rheological properties of PC/ABS polymer blends from the perspective of relaxation behavior and the formation and breakdown of physical structure at various the shear conditions. The viscoelastic properties of PC/ABS blends were examined in terms of measuring temperature and blend composition.

## 2. Materials and Methods

### 2.1. Materials

The main components of PC/ABS blends, PC (LUPOY GP1000MU, LG chemical, Daejeon, Korea), styrene-co-acrylonitrile and PBD with average diameters of 300 nm (grafting degree of SAN:0.15), were purchased from LG chemical, Daejeon, Korea. All materials were dried at 80 °C overnight in a vacuum oven prior to use. PC/ABS blends were prepared by a melt compounding process using a twin-screw extruder (Hankook EM, STS32H, Pyoungtaek, Korea). PC, SAN and PBD were mixed by tumbling in a polyethylene bag and the mixture was placed in a hopper. The processing temperature was between 220 °C and 250 °C from the hopper to the die and the mixing screw speed and feeding rate were 250 rpm and 45 kg/hr, respectively. The blend ratios of PC/ABS were 10:0, 9:1, 7:3, 5:5, 3:7, 1:9, 0:10 and the detailed compositions of PC, SAN and PBD are listed in Table 1. Disk-shaped specimens were compression molded at 250 °C for 3 min for the measurement of rheological properties of PC/ABS polymer blends.

### 2.2. Physical Property Measurement

An advanced Rheometric Expansion System (ARES G2, TA Instruments, New Castle, DE, USA) was used to analyze chain motion behavior with regard to the rheology. Specimens with diameters of 25 mm were used with a parallel-plate geometry. The plate gap was 2 mm and excess sample during gap setting was removed before the test. The specimens were held for 5 min between the plates at a given temperature to eliminate thermal history and residual stress prior to the measurements. The frequency sweep measurement was implemented over the angular frequency range of 0.01 to 500 rad/s at 240, 250 and 260 °C with a 7% strain condition. The time dependence of PC/ABS blends was measured using a time sweep test mode for 2400 s at 250 °C with given angular frequency levels of 0.1, 1 and 10 rad/s. In addition, the sequence test mode was designed to investigate the shear dependent response of PC/ABS blends. At first, a frequency sweep test was carried out over the frequency range of 0.05 to 200 rad/s and then a strain sweep test over a strain level of 0.1 to 1000% under a constant frequency of 1 rad/s was conducted. The strain sweep measurement was repeated 3 times and the frequency sweep measurement was conducted again. In summary, a sequence test mode consisted of a first run of frequency sweep run, 3 repetitive strain sweep runs and a second frequency sweep run. The morphology of PC/ABS blend was examined by using transmission electron microscopy (TEM; JEOL, JEM-2100F, Tokyo, Japan). The ultra-microtomed section was stained with OsO_4_ for 30 s in a vacuum to cause the PBD in the SAN matrix to turn black and then, the PC phase was stained with RuO_4_ for 15 min, resulting in a grey appearance in the TEM pictures.

## 3. Results and Discussion

As the angular frequency increased for the case of dynamic oscillatory shear, the viscosities of all PC/ABS blends decreased because thermoplastic polymer chains generally show shear-thinning behavior with increasing shear rate. These phenomena were observed over the entire range of measurement temperatures and the change in viscosity was more remarkable at larger amount of ABS. However, pure PC polymer did not show significant change in viscosity over the entire frequency range. At low shear conditions, the viscosities of PC/ABS blends exhibited higher values as the amount of ABS enlarged. The viscosity of the 3:7 (PC:ABS) blend ratio was higher than even that of pure ABS. In general, melted polymer chains gradually decreased in the viscosity as temperature increased because high temperature results in high chain mobility. At the low shear levels of 0.01 rad/s, however, the viscosities of ABS-rich blends raised at higher measurement temperature. Changes in viscosities of PC/ABS blends at various temperatures are depicted in Figure 1 and Appendix A.

The effects of temperature on the viscosity of each PC/ABS blend at a given frequency are depicted in Figure 2. At the frequencies exceeding 0.1 rad/s, all PC/ABS blends exhibited decreases in viscosity with an increase in measurement temperature, which indicates the general flow behavior in accordance with temperature. At the extremely low frequency level of 0.01 rad/s, however, the viscosities of ABS-rich blends increased as temperature got higher except for pure PC polymer and the PC-rich 9:1 (PC:ABS) blend. This phenomenon is most apparent at a blend ratio of 3:7 (PC:ABS) and the viscosity did not change regardless of measurement temperature even at higher frequency levels of 0.1 rad/s, where the other blends showed lesser viscosities at higher temperature. At low shear conditions where the material characteristics prevail over the processing characteristics, the internal structures maintain their initial states when the applied shear stress is not sufficient to cause critical shear stress. This indicates that high chain mobility induced by high temperatures plays a major role in the formation of a network structure for ABS-rich blends, rather than free chain motion, which can cause the breakdown of the structure. Figure 1; Figure 2 show that PC and ABS polymers at a blend ratio of 3:7 (PC:ABS) build internal structure under sufficiently low shear conditions and the collapse of the structure occurs when sufficient external force exceeding the critical stress is applied.

To investigate the effects of shear conditions on the PC/ABS blends, time sweep measurements were carried out at a given frequency. The complex viscosity, η*(t), was divided by its initial viscosity, η*(0) and this normalized viscosity offers time-dependent rheological responses such as chain orientation, stability and network formation, among other aspects [47,48,49]. Time dependence of the normalized viscosity of PC/ABS blends at a given frequency is plotted in Figure 3. The specimens with more than 30 wt% ABS exhibited increases in viscosity at low frequency levels of 0.1 rad/s as opposed to high frequency. Unlike the other specimens, pure PC polymer exhibited gradual increases in normalized viscosity at high frequency over time. This is because PC polymer chains go through a transition to an ordered structure at high frequency [50]. In particular, the viscosity at the blend ratio of 9:1 showed no specific changes with frequency, which demonstrates that the structural transition of PC chains and the building and breakdown of internal network structure of ABS occur simultaneously in the blend systems. Thus, viscosities of PC/ABS blends with more than 30 wt% ABS increased at low frequency, where network formation is more dominant than breakdown because the frequency level of 0.1 rad/s is not sufficient to influence the internal structure. The shear dependent phase change behaviors examined by TEM were exhibited in Appendix A.

Conventional rheological measurement techniques have employed small-amplitude oscillation shear (SAOS) analysis due to its robust theoretical basis and the resulting linear rheological responses. Recently, many studies have focused on the large-amplitude oscillation shear (LAOS) analysis technique to analyze the non-linear behavior of materials [40,41,42,43,44,45,46]. LAOS analysis has been widely used to analyze complex systems because the materials respond more sensitively to the external stimuli in large strain shear condition. In this study, the LAOS analysis method was modified to apply a sequence test mode to repeatedly perform SAOS and LAOS and the results are provided in Figure 4 and Figure 5. First, frequency sweep tests with 7% of strain were carried out. Immediately thereafter, large strain sweep tests with strain ranges from 0.1 to 1000% were repeated three times. Then, the frequency sweep test was performed again to investigate the decrease in viscosity before and after the large strain amplitude measurement. These tests showed that the change in viscosity after 3 large strain amplitude mode runs was not large for homogenous PC but the viscosity drop was more noticeable as the amount of ABS increased. In particular, the viscosity at the low frequency level of 0.1 rad/s and the change in viscosity of the 3:7 blend ratio were greater than those of pure ABS. This suggests insufficient recovery of the structure from external stress due to a longer relaxation time in the polymer blend during sequence mode measurement. Intermolecular interactions between PC, SAN and PBD play a significant role in relation to the chain relaxation behavior. Chun et al. [51] investigated the miscibility of PC, SAN and PBD. They approached this problem in terms of differences in solubility parameter, Δδ, using a group contribution method. The Δδ between PC and SAN was 0.20 (cal cm^−3^)^1/2^, whereas that between PC and PBD was 1.85 (cal cm^−3^)^1/2^, indicating that PC and SAN exhibited higher compatibility than PC and PBD. Similar results were found by Aid et al. [52].

In addition, PC and SAN in a PC/SAN/PBD tertiary system, formed needle-like co-continuous structures where PBD was dispersed in a SAN matrix in a sufficient shear field, whereas ABS formed a sea-island structure in which PBD was dispersed in SAN [17,18,19]. That is, PC/ABS polymer blend had a pseudo-structure attributed to the intermolecular interactions between PC and SAN, where PBD was dispersed in a SAN matrix. This pseudo-structure formed by intermolecular interactions between PC and SAN resulted in higher viscosity at a blend ratio of 3:7 than that of the pure ABS. Hyun et al. [40]. classified the change of internal structure according to strain amplitude into four types—strain thinning, strain hardening, weak strain overshoot and strong strain overshoot. As the ABS content increased, the overshooting phenomenon (in which G′ and G″ increased followed by decreasing) was observed, which indicates that the strength of the microstructure was enhanced at high strain amplitude conditions. Consequently, the formation of internal structure ascribed to the intermolecular interactions between PC and SAN and the presence of PBD were the dominant factors influencing the critical yield stress between PC and ABS at low frequency or low strain shear conditions. However, at high frequency or high strain shear conditions above yield point, the co-continuous structure and the amount of PBD were dominant effect, leading to the strongest structure.

To understand this unusual behavior at low frequency in detail, the yield stresses for all PC/ABS polymer blends were measured at various temperatures as shown in Figure 6 and Appendix A. The strength of the physical structures can be quantified by the value of the yield stress, which indicates the minimum energy required to break down the physical structures. The yield stress can be expressed by a Casson plot defined by Equation (1) [53,54,55]:*G*″^1/2^ = *G*_y_″^1/2^+ *Kω*^1/2^(1)

Here, *G*_y_″ denotes the yield stress, *K* is a constant and *ω* is the frequency. As expected from the results of the sequence mode measurement, the yield stress exhibited the highest value at a PC/ABS polymer blend ratio of 3:7. This indicates that the strongest physical structure was formed at a blend ratio of 3:7 among all PC/ABS polymer blends. Previously, we observed that the viscosity of the blends increased with increasing temperature at low shear conditions. Likewise the yield stress at higher temperature exhibited growth up. On the basis of rheological principles, a higher temperature entails a drop in viscosity and a lower resistance to flow because it causes higher chain mobility. In the PC/ABS polymer blend system, however, the increased chain mobility results in the formation of internal structure, which leads to an increase in viscosity under low shear conditions. Moreover, the intermolecular interactions between PC and SAN have a synergetic effect on the build-up of internal structure. Hence, the most balanced structure is generated at a blend ratio of 3:7 rather than for pure ABS even with the higher PBD content.

The presence of physical structure is expected to influence the relaxation behavior of polymeric systems. That is, if polymer chain mobility is hindered, then the response to the external stimuli will be delayed. The relaxation time (λ) was calculated using an empirical equation as follows [53,54,55].
*λ* = *G*′/(|*η**| *ω*^2^)(2)

The relaxation time would be longer if molecular order or internal structures are present. As described in Figure 7 and listed in Table 2, the pure PC polymer recorded the shortest relaxation time because PC chains did not undergo the formation of network structure accompanying elasticity. High temperature results in an increase in chain mobility rather than formation of internal structure in the PC, so the relaxation time decreased with increasing temperature. However, as the content of ABS increased, the relaxation time increased at higher temperature. As in the previous results from the sequence mode test, the relaxation time, a parameter that can determine the strength of the network structure, was highest at the blend ratio of 3:7 at low shear, whereas pure ABS with a high content of PBD exhibited the longest relaxation time at high shear when the internal structure collapsed. In conclusion, the internal structure formed by PC and ABS becomes gradually stronger with increasing temperature at low shear but when sufficient external force above yield stress is applied, the collapse of the structure is accelerated by temperature and relaxation time is also reduced.

The logarithmic plot of storage modulus (G′) versus loss modulus (G″), that is, the so-called modified Cole-Cole plot, is depicted in Figure 8 and the initial slopes at the various blend ratios and measurement temperatures are listed in Table 3. A slope of 2 in the plot represents an isotropic and homogeneous system without any internal structure or specific intermolecular interactions [22,23,24,25,26]. As the content of ABS increases, the slopes of the curves decreased. In other words, the heterogeneity of the systems increased. PC/ABS blend ratios of 3:7, 1:9 and 0:10 showed similar responses to the applied shear stress, whereas the others showed a decreasing slope in order of blend ratio. It is noticeable that the heterogeneity of pure PC and the polymer blend with a ratio of 9:1 decreased with increasing temperature, whereas specimens with more than 30 wt% of ABS became more heterogeneous at high temperature.

## 4. Conclusions

The effects of temperature and blend composition on the rheological response of PC/ABS polymer blends were investigated in terms of their internal physical structure and chain relaxation behavior. In low shear fields, the viscosity of the blend ratio of 3:7 (PC:ABS) was higher than that of pure ABS (which has much higher PBD contents) because of the internal structure formed by intermolecular interactions between PC and ABS. High temperature generally enhances the chain mobility leading to a decrease in viscosity. ABS-rich blends, however, exhibited an increase in viscosity at low shear conditions with high measurement temperatures. That is, higher chain mobility resulted in pseudo-structure formation rather than flowability. This phenomenon was the most significant at the blend ratio of 3:7. The critical yield stress between PC and ABS was influenced by the formation of a physical structure because of the intermolecular interactions between PC and SAN are dominant at low shear conditions, whereas a co-continuous phase and the presence of PBD become dominant at high shear conditions. Among all specimens, the yield stress showed the highest value at a blend ratio of 3:7 and the yield stress of ABS-rich blends increased as measurement temperature increased due to the internal structure. A stronger internal structure resulted in a longer relaxation time and this result agreed well with the increase in viscosity or yield stress in the low shear region. Results from the relaxation time, the internal structure formed by PC and ABS became gradually stronger with increasing temperature at low shear but when sufficient external force above the yield stress was applied, the structural collapse was accelerated by temperature.

## Figures and Tables

**Figure 1 polymers-12-01916-f001:**
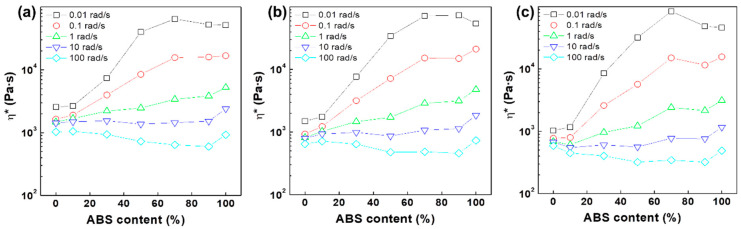
Change of viscosity of PC/ABS blends at given frequencies; measured at (**a**) 240, (**b**) 250 and (**c**) 260 °C.

**Figure 2 polymers-12-01916-f002:**
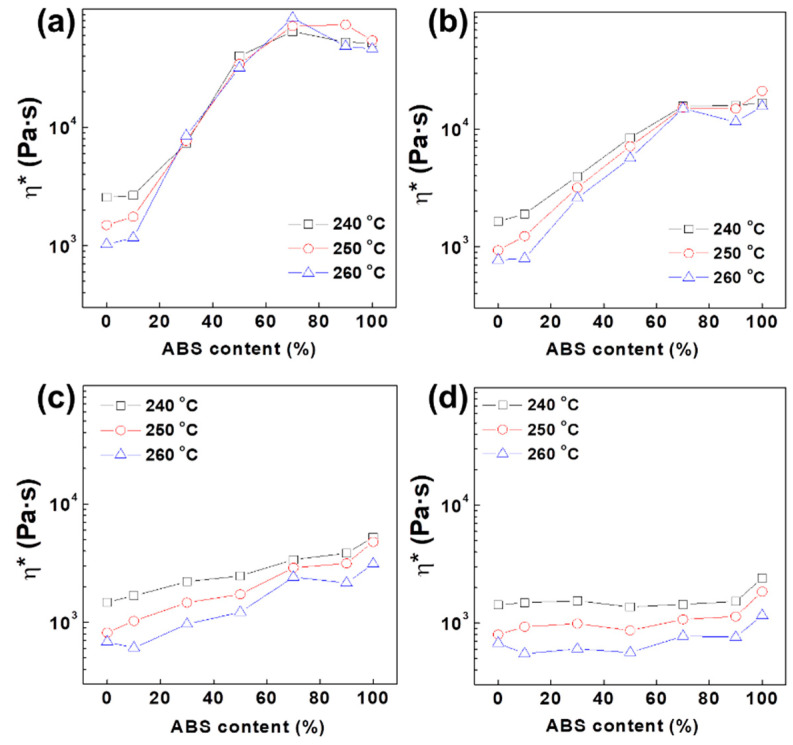
Effect of temperature on viscosity of PC/ABS blends at given frequency of (**a**) 0.01, (**b**) 0.1, (**c**) 1 and (**d**) 10 rad/s.

**Figure 3 polymers-12-01916-f003:**
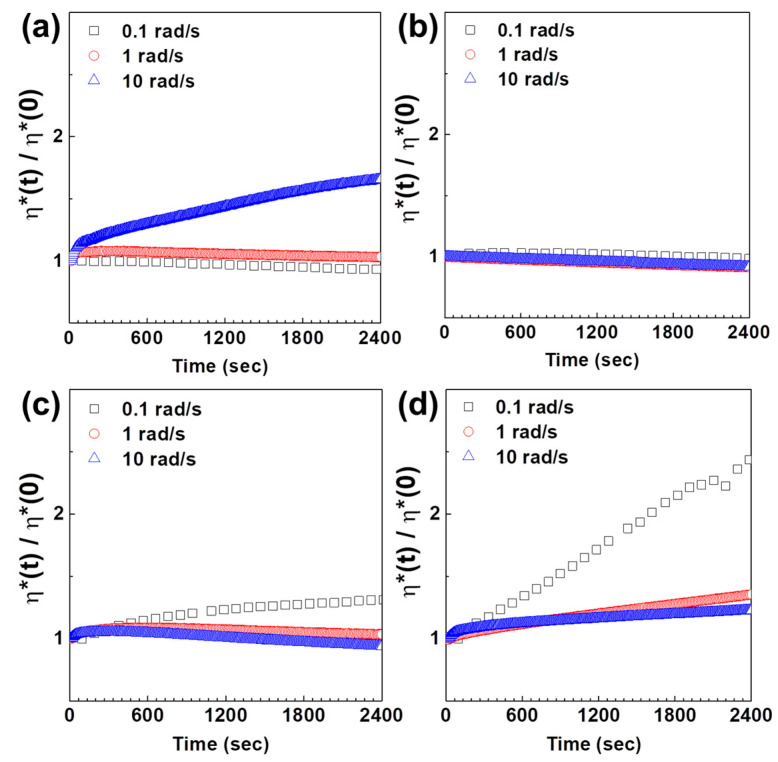
Time dependence of the normalized viscosity (η^*^(t)/η^*^(0)) of PC/ABS with blend ratio of (**a**) 10:0, (**b**) 9:1, (**c**) 7:3 and (**d**) 0:10 at constant frequencies.

**Figure 4 polymers-12-01916-f004:**
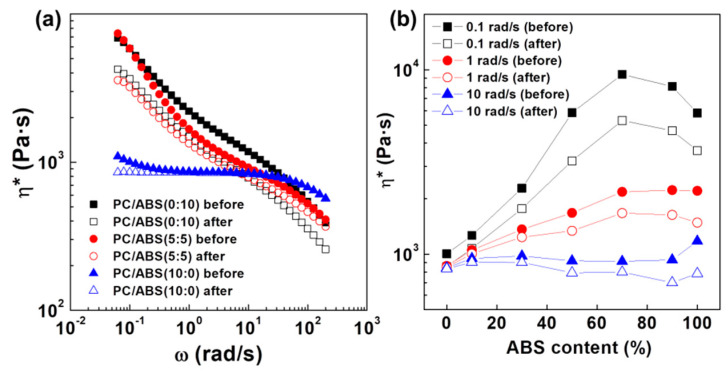
Change of viscosity of PC/ABS blends before and after strain sweep applied sequence mode; (**a**) viscosity curve and (**b**) change of complex viscosity with increasing ABS content.

**Figure 5 polymers-12-01916-f005:**
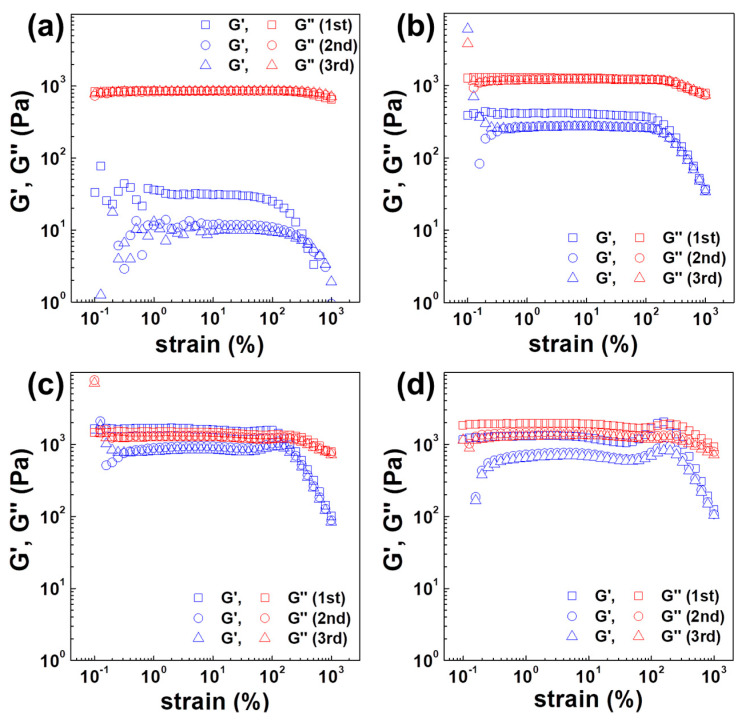
Change of storage and loss moduli during strain sweep test for PC/ABS blend ratio of (**a**) 10:0, (**b**) 7:3, (**c**) 3:7 and (**d**) 0:10.

**Figure 6 polymers-12-01916-f006:**
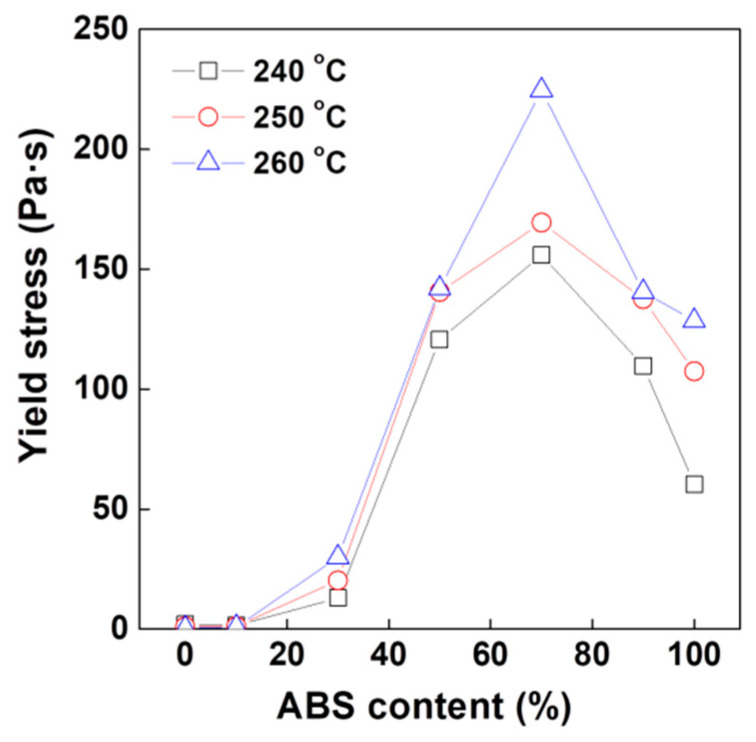
Yield stress (G_y_″) of PC/ABS blend with increasing ABS content at given measuring temperature.

**Figure 7 polymers-12-01916-f007:**
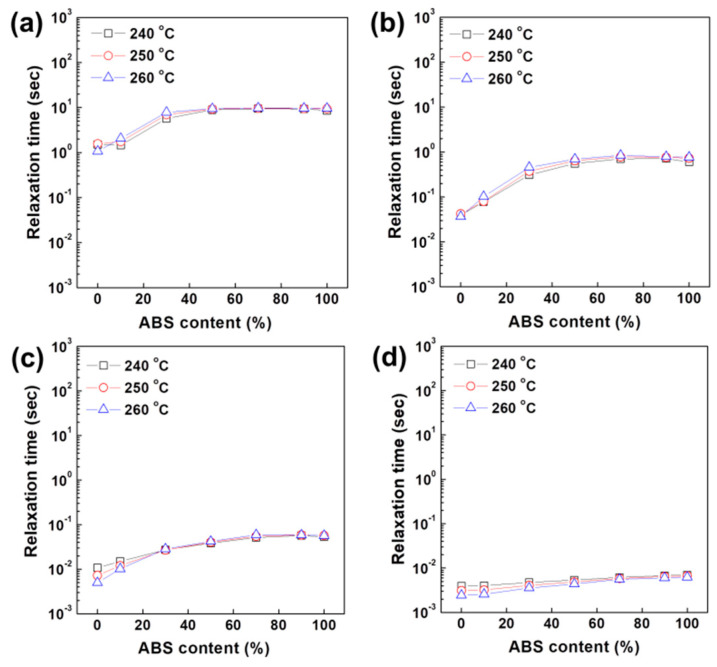
Change of relaxation time of PC/ABS blends at given frequency of (**a**) 0.1, (**b**) 1, (**c**) 10 and (**d**) 100 rad/s.

**Figure 8 polymers-12-01916-f008:**
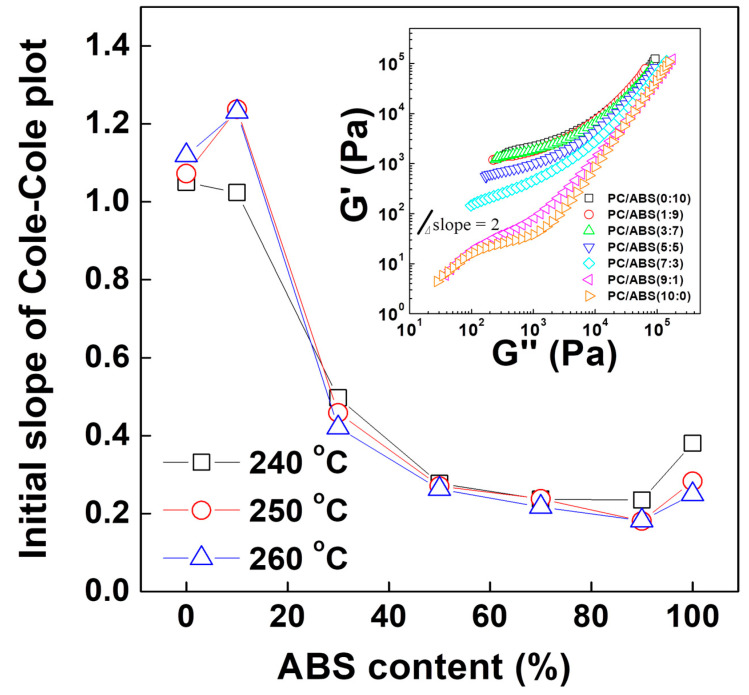
Initial slope of modified Cole-Cole plot at PC/ABS blend with increasing ABS content at given temperatures.

**Table 1 polymers-12-01916-t001:** Blend composition of polycarbonate/acrylonitrile-butadiene-styrene (PC/ABS).

Blend Ratio (PC:ABS)	10:0	9:1	7:3	5:5	3:7	1:9	0:10
PC (wt%)	100	90	70	50	30	10	0
PBD (wt%)	0	4	12	20	28	36	40
SAN (wt%)	0	6	18	30	42	54	60

**Table 2 polymers-12-01916-t002:** Relaxation time of PC/ABS blends at various temperatures.

Temp. (°C)	Frequency (rad/s)	Relaxation Time (s)						
		10:0	9:1	7:3	5:5	3:7	1:9	0:10
240	0.1	1.52	1.45	5.73	8.91	9.41	9.30	8.60
	1	0.04	0.078	0.31	0.56	0.7	0.71	0.6
	10	0.011	0.015	0.027	0.039	0.051	0.057	0.053
	100	0.004	0.004	0.0048	0.0054	0.0062	0.0068	0.007
250	0.1	1.58	1.74	6.97	9.25	9.62	9.51	9.44
	1	0.042	0.08	0.37	0.64	0.78	0.77	0.73
	10	0.0073	0.012	0.027	0.041	0.055	0.06	0.057
	100	0.0031	0.0032	0.0041	0.0049	0.0058	0.0065	0.0067
260	0.1	1.07	2.06	7.85	9.41	9.75	9.65	9.59
	1	0.037	0.1	0.46	0.7	0.84	0.79	0.77
	10	0.005	0.01	0.028	0.042	0.59	0.058	0.057
	100	0.0025	0.0026	0.0036	0.0044	0.0056	0.006	0.0063

**Table 3 polymers-12-01916-t003:** Initial slopes of modified Cole-Cole plot of PC/ABS blends.

Temperature (°C)	Initial Slope of Modified Cole-Cole Plot						
	10:0	9:1	7:3	5:5	3:7	1:9	0:10
240	1.05	1.02	0.50	0.28	0.24	0.23	0.38
250	1.07	1.24	0.46	0.27	0.24	0.18	0.28
260	1.12	1.23	0.42	0.26	0.22	0.18	0.25

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
