# Peer review of "Rheological Investigation of Relaxation Behavior of Polycarbonate/Acrylonitrile-Butadiene-Styrene Blends"

_polymers, 2020, doi:10.3390/polym12091916_

Round 1
Reviewer 1 Report
The manuscript by Seo et al. describes the rheological behaviors of PC/ABS blends. The topic is meaningful. However, it is hard to catch the theoretical advances of this work compared with previous researches. Thus I recommend major revision of this manuscript before publication. Detailed comments are listed below.
- It is better for the authors to highlight the challenges in the area and their new findings in the instruction.
- The authors need to justify the temperatures they used in the study. In my opinion, three temperatures may not enough. Besides, the gap of only 20°C is also too small. The information delivered under those temperatures is too similar to each other.
3. It would be better if the authors can increase the resolution of the figures.
Author Response
Comments from Reviewer #1
The manuscript by Seo et al. describes the rheological behaviors of PC/ABS blends. The topic is meaningful. However, it is hard to catch the theoretical advances of this work compared with previous researches. Thus I recommend major revision of this manuscript before publication. Detailed comments are listed below.
- It is better for the authors to highlight the challenges in the area and their new findings in the instruction.
Our Response: Thank you so much for the comment. One of the important highlights is the unusual phenomenon in which the viscosity increases as temperature increases at low shear condition. On the basis of rheology, high temperature brings about the lesser viscosity due to increased chain mobility which leads to better flowability. However, we found that the enhanced chain mobility plays a role in network formation rather than free chain motion for the ABS-rich blends. And this is more noticeable at the blend ratio of 3:7, in which the critical shear stress recorded the highest value among other blends. That is, the strongest internal structure was formed in the blend ratio of 3:7. We updated the paragraph as follows.
(Added in manuscript, 4-126)
At low shear conditions where the material characteristics prevail over the processing characteristics, the internal structures maintain their initial states when the applied shear stress is not sufficient to cause critical shear stress. This indicates that high chain mobility induced by high temperatures plays a major role in the formation of a network structure for ABS-rich blends, rather than free chain motion, which can cause the breakdown of the structure.
- The authors need to justify the temperatures they used in the study. In my opinion, three temperatures may not enough. Besides, the gap of only 20°C is also too small. The information delivered under those temperatures is too similar to each other.
Our Response: In industrial fields, common processing temperature is about 300 oC for PC and 220 oC for ABS. There is a large temperature gap between amorphous PC and ABS, and when the melt blending using these two materials is performed, the moderate temperature about 250 oC is recommended for preventing thermal degradation of ABS and underplasticizing of PC. Thus, many researches dealing with the rheological and viscoelastic properties of PC/ABS were carried out in the temperature range from 230 to 260 oC. We do agree that the rheological properties can be investigated more clearly with larger temperature gap. However, we obtained the meaningful results at the temperature range of 20 oC, in other words, the viscoelastic properties of PC/ABS blends are significantly sensitive even in the small range of 20 oC. In addition, the results from other researches on the rheological properties of PC/ABS with the temperature gap of 10 or 20 oC exhibited the difference in the degree of viscoelasticity, but did not show notable changes such as phase changes. Therefore, we would like to give more significance in the analysis of the rheological behaviors that can be distinguished even in a narrower temperature range. And we expect that our results will have a great impact on the industrial fields when published. Because when a product is manufactured by injection molding, a temperature deviation can occur even if the temperature of the machine is set as constant especially for the products with large and complex shapes such as automotive parts. Here, the results that the viscoelastic properties of PC/ABS blend can be changed sensitively according to the blend ratio with small temperature range and shear condition can suggest a new trouble shooting guide.
References in main text and response letter
- Boronat, T.; Segui, V.J.; Peydro, M. A.; Reig, M.J. Influence of temperature and shear rate on the rheology and processability of reprocessed ABS in injection molding process. J Mater Process Technol, 2009, 209,
- Balakrishnan, S.; Neelakantan, N.R.; Saheb, D.N.; Jog, J.P. Rheological and morphological behaviour of blends of polycarbonate with unmodified and maleic anhydride grafted ABS. Polymer, 1998, 39, 5765-5771.
- Babbar, I.; Mathur, G.N. Rheological properties of blends of polycarbonate with poly(acrylonitrile-butadiene-styrene). Polymer, 1994, 35, 2631-2635.
(Newly added references)
- Rostami, A.; Masoomi, M.; Fayazi, M.J.; Vahdati, M. Role of multiwalled carbon nanotubes (MWCNTs) on rheological, thermal and electrical properties of PC/ABS blend. RSC Advances, 2015, 5(41), 32880-32890.
- Khan, M. M. K.; Liang, R. F.; Gupta, R. K.; Agarwal, S. Rheological and mechanical properties of ABS/PC blends. Korea-Aust Rheol J, 2005, 17(1), 1-7.
- It would be better if the authors can increase the resolution of the figures.
Our Response: The figure resolutions are modified as 300 dpi in revised manuscript.

Reviewer 2 Report
Interesting manuscript where the relaxation behavior of PC/ABS blends is reported. However, before acceptance, some issues need to be addressed:
- Too many decrease/increase expressions, alternative words should be used
- Page 2-62. The objective proposed in the introduction is not demonstrated in the paper, there is the need of additional experimental techniques to support the objective.
- The authors indicate in page 4-132 that a single measurement can lead to different molecular explanations. However, in many sentences, hypotheses are taken on the basis of rheological measurements alone, some examples: page 5-137; 160; 170, page 7-203, page 8-222
- The conclusions section is a summary of the previous claims. It is also too long and should be shortened, indicating only the most important facts and molecular conclusions. Experimental evidences should be shown, or at least test a couple, when citations to the work of others are used.
Author Response
Comments from Reviewer #2
Interesting manuscript where the relaxation behavior of PC/ABS blends is reported. However, before acceptance, some issues need to be addressed:
- Too many decrease/increase expressions, alternative words should be used
Our Response: The decrease/increase expressions were replaced by adequate diverse words in revised manuscript.
(Modified expressions in manuscript, colored blue)
3-106, 109, 112, 122, 6-193, 7-216
- Page 2-62. The objective proposed in the introduction is not demonstrated in the paper, there is the need of additional experimental techniques to support the objective.
Our Response: We investigated the rheological properties of PC/ABS blend with regard to the chain motions analyzed by newly designed sequence mode in large strain sweep, the formation and breakdown of internal structures originated from the relation among PC, SAN, PBD. The physical strength of the structure was established by yield stress obtained from Casson plot. As reviewer #2 mentioned, we found the shear-dependent phase change for PC/ABS blend by TEM analysis. The phase of the blend could be distinguished by staining technique, OsO4 for PBD and RuO4 for PC. In Figure S1, the morphology formation according to the shear condition was depicted for the case of PC/ABS blend and that of pure ABS. Here, we observed the difference in phase change behavior in terms of relations between PC and SAN phases.
Figure S1. The phase change of PC/ABS blends examined by TEM for the blend ratio of (a-d) 5:5 and (e-h) 0:10; quenched to the room temperate right after time sweep measurement with the frequency of (a-b) 0.1 rad/s, (c-d) 1 rad/s, and the images of (e-h) are obtained from the same corresponding conditions.
- The authors indicate in page 4-132 that a single measurement can lead to different molecular explanations. However, in many sentences, hypotheses are taken on the basis of rheological measurements alone, some examples: page 5-137; 160; 170, page 7-203, page 8-222
Our Response: The rheological approach to analyze chain motion has been widely used by many researchers. As reviewer mentioned, a single measurement can be applied to the explanation of molecular level of deformation, however, it did not mean that one single rheological test method can be a universal solution for the explanation. We only adopted the methods already proven by others or cited the similar results from other researchers. Our opinions to the sentences pointed out are as follows.
4-132 : application of η*(t)/η*(0) parameter is mentioned in ref. 47-49.
References in main text and response letter
- 47. Chae, D.W.; Lim, J.H.; Seo, J.S.; Kim, B.C. Variation of physical properties of nylon-66/clay nanocomposites with preparation conditions. Korea-Aust Rheol J, 2012, 24, 45-52.
- 48. Chae, D.W.; Nam, Y.; An, S.G.; Cho, C.G.; Lee, E.J.; Kim, B.C. Effects of molecular architecture on the rheological and physical properties of polycaprolactone. Korea-Aust Rheol J, 2017, 29, 129-135.
- 49. Bae, W.-S.; Lee, S.; Kim, B.C. Effect of shear condition on the thermal stabilization of ethylene–propylene–carbon monoxide terpolymer. Polym Degrad Stabil, 2014, 105, 160-165.
5-137 : The phase transition of PC chain to ordered structure with increasing frequency is mentioned in ref. 50.
References in main text and response letter
- Memon, N.A. Rheological properties and the interface in polycarbonate/impact modifier blends: Effect of modifier shell molecular weight. J Polym Sci Part B: Polym Phys, 1998, 36, 1095-1105.
5-160 : The inter/intra-molecular interactions significantly affect the chain motions. The difference in intermolecular interactions among three components, PC, SAN, and PBD, were explained by following paragraph in terms of solubility parameters.
5-170 : The morphology formation of PC and ABS has been investigated many researchers and the results were described in ref. 17-19. Our results also exhibit similar phase changes depicted in Figure S1.
References in main text and response letter
- 17. Lee, M.-P.; Hiltner, A.; Baer, E. Formation and break-up of a bead-and-string structure during injection moulding of a polycarbonate/acrylonitrile-butadiene-styrene blend. Polymer, 1992, 33, 675-684.
- 18. Lee, M.-P.; Hiltner, A.; Baer, E. Phase morphology of injection-moulded polycarbonate/acrylonitrile-butadiene-styrene blends. Polymer, 1992, 33, 685-697.
- 19. Memon,A. Rheological properties and the interface in polycarbonate/impact modifier blends: Effect of modifier shell molecular weight. J Polym Sci Part B: Polym Phys, 1998, 36, 1095-1105.
7-203 : High temperature gives polymer chains high mobility to freely rotate, and disentanglement of the chains gradually proceeds as temperature increases. This phenomenon leads to the decrease in viscosity.
8-222 : Unlike ABS, PC is a homogeneous polymer and it did not show notable elastic behavior due to the lack of PBD. In addition, it exhibited only general viscoelastic behavior as temperature increases. We already explained the effects of high temperature on the chain mobility, which brings about the lesser viscosity. PC clearly follows this rule, whereas high temperature greatly contributed to the formation of network structure for the ABS-rich blends. This is the reason that PC showed the shortest relaxation time.
- The conclusions section is a summary of the previous claims. It is also too long and should be shortened, indicating only the most important facts and molecular conclusions. Experimental evidences should be shown, or at least test a couple, when citations to the work of others are used.
Our Response: We revised the conclusion paragraph shorter and simpler with the significant results.

Reviewer 3 Report
The submitted manuscript reports about a detailed rheological study of PC/ABS blends.
In my view, the aim of the study is not clearly stated and the discussion of obtained results is confused. Furthermore, the following concerns should be solved:
- The Abstract should be modified in order to make explicit the aim of the study.
- In the Introduction, some sentences and related references about the typical rheological behavior of immiscible polymer blends should be added.
- Page 3, line 98 the Authors stated "...for the case of steady shear..." referring to data in Figure 1, which reports result of oscillatory tests. Please, clarify this issue.
- The whole viscosity curve as a function of frequency for all investigated systems should be reported instead of single points at different frequencies. in fact, the trend of viscosity over the tewsted frequency interval is very important to fully characterize the rheological response of polymer-based systems.
- the yield stress values have been calculated through Casson model, which is usually used to model data coming from steady state measurements. In the manuscript, this model was applied to data from oscillatory measurements, therefore the Authors should verify its applicability and validity.
- A reference should be added to support the empirical rule used to calculate the relaxation time.
Author Response
Comments from Reviewer #3
The submitted manuscript reports about a detailed rheological study of PC/ABS blends.
In my view, the aim of the study is not clearly stated and the discussion of obtained results is confused. Furthermore, the following concerns should be solved:
- The Abstract should be modified in order to make explicit the aim of the study.
Our Response: Thank you for your kind advice. We modified the abstract session to express the objective and significance of our research.
(Modified in manuscript, 1-13)
The rheological properties of polycarbonate/acrylonitrile-butadiene-styrene (PC/ABS) blends with various blend ratios are investigated at different temperatures to determine the shear dependent chain motions in heterogeneous blend system.
- In the Introduction, some sentences and related references about the typical rheological behavior of immiscible polymer blends should be added.
Our Response: The following sentences are added in manuscript.
(Added in manuscript, 2-57)
The rheological properties of the immiscible polymer blends systems are closely related to the morphology formation and compatibility between the blend components.[00] Because the rheological responses are systematical and macroscopic, little differences in molecular motions can bring about different viscoelastic properties.
(Newly added references)
- Martin, P.; Carreau, P.J.; Favis, B.D.; Jérôme, R. Investigating the morphology/rheology interrelationships in immiscible polymer blends. J Rheol, 2000, 44(3), 569-583.
- Kitade, S.; Ichikawa, A.; Imura, N.; Takahashi, Y.; Noda, I. Rheological properties and domain structures of immiscible polymer blends under steady and oscillatory shear flows. J Rheol, 1997, 41(5), 1039-1060.
- Wang, H.; Yang, X.; Fu, Z.; Zhao, X.; Li, Y.; Li, J. Rheology of nanosilica-compatibilized immiscible polymer blends: Formation of a “heterogeneous network” facilitated by interfacially anchored hybrid nanosilica. Macromolecules, 2017, 50(23), 9494-9506.
- Page 3, line 98 the Authors stated "...for the case of steady shear..." referring to data in Figure 1, which reports result of oscillatory tests. Please, clarify this issue.
Our Response: We just used the word in accordance with general shear flow, however, as reviewer #3 pointed out, this shear field should be corrected precisely as oscillatory test.
(Modified in manuscript, 3-103)
... for the case of dynamic oscillatory shear, ...
- The whole viscosity curve as a function of frequency for all investigated systems should be reported instead of single points at different frequencies. in fact, the trend of viscosity over the tested frequency interval is very important to fully characterize the rheological response of polymer-based systems.
Our Response: We absolutely agree with reviewer’s opinion. We just only re-plotted the specific points to compare the difference in viscosity behavior in detail because it is difficult to fine the difference between 7 kinds of specimens in a whole viscosity curves as temperature increases. However, we think the trend of viscosity from the viscosity curve is the most significant to analyze the basic shear dependent chain motion as reviewer mentioned. We depicted the viscosity curves at give temperatures as follows in Figure S2.
Figure S2. Viscosity curves of PC/ABS blends at the measuring temperature of (a) 240 oC, (b) 250 oC, and (c) 260 oC.
- The yield stress values have been calculated through Casson model, which is usually used to model data coming from steady state measurements. In the manuscript, this model was applied to data from oscillatory measurements, therefore the Authors should verify its applicability and validity.
Our Response: The Casson model can be applied in dynamic oscillatory measurements by extrapolation of initial loss modulus regions. The intercept of the plot can be applied to the yield stress as described in ref. 48-50. The whole curve and detailed extrapolation method, the change of yield stress are exhibited in Figure S3.
References in main text and response letter
- 53. Bae, W.-S.; Kwon, O.J.; Kim, B.C.; Chae, D.W. Effects of multi-walled carbon nanotubes on rheological and physical properties of polyamide-based thermoplastic elastomers. Korea-Aust Rheol J, 2012, 24, 221-227.
- 54. Bae, J.; Lee, S.; Kim, B.C.; Cho, H.H.; Chae, D.W. Polyester-based thermoplastic elastomer/MWNT composites: Rheological, thermal, and electrical properties. Fiber Polym, 2013, 14, 729-735.
- 55. Eom, Y.; Jung, D.E.; Hwang, S.S.; Kim, B.C. Characteristic dynamic rheological responses of nematic poly(p-phenylene terephthalamide) and cholesteric hydroxypropyl cellulose phases. Polym J, 2016, 48, 869-874.
Figure S3. The yield stress behavior of PC/ABS blends; (a) Casson plot of the blends with whole frequency range at 250 oC, (b) difference in intercepts of the blends, (c) change of yield stress values of the blends with increasing ABS content at given temperatures.
- A reference should be added to support the empirical rule used to calculate the relaxation time.
Our Response: The references regarding the relaxation time are included in the modified manuscript.
Round 2
Reviewer 1 Report
The issues that I raised in the previous review have been addressed. I think this paper can be accepted in Polymers now.
Reviewer 2 Report
Accept as it is
Reviewer 3 Report
I suggest the publication of the manuscript as it stands, as the Authors modified it following the suggestion of the Reviewer